# ICU: Conquering Language Barriers in Vision-and-Language Modeling by Dividing the Tasks into Image Captioning and Language Understanding

**Guojun Wu**

Department of Computational Linguistics, University of Zurich

guojun.wu@uzh.ch

## Abstract

Most multilingual vision-and-language (V&L) research aims to accomplish multilingual and multimodal capabilities within one model. However, the scarcity of multilingual captions for images has hindered the development. To overcome this obstacle, we propose ICU[1], Image Caption Understanding, which divides a V&L task into two stages: a V&L model performs image captioning in English, and a multilingual language model (mLM), in turn, takes the caption as the alt text and performs cross-lingual language understanding. The burden of multilingual processing is lifted off V&L model and placed on mLM. Since the multilingual text data is relatively of higher abundance and quality, ICU can facilitate the conquering of language barriers for V&L models. In experiments on two tasks across 9 languages in the IGLUE benchmark, we show that ICU can achieve new state-of-the-art results for five languages, and comparable results for the rest.

## 1 Introduction

In recent times, there has been a growing interest in extending the success of vision-and-language (V&L) models beyond English to encompass non-English languages. However, the scarcity of training data has posed challenges in the development of multilingual models. To address this issue, various code-switch strategies (Ni et al., 2021; Nooralahzadeh and Sennrich, 2022) have been proposed to encourage models to learn the relationships between corresponding words in different languages. Additionally, machine translation (MT) techniques have been employed to augment existing English-only datasets (Qiu et al., 2022; Zhou et al., 2021). Although some improvements have been achieved using MT-enhanced translated data, the quality of translations varies across languages. Furthermore, fine-tuning strategies (Liu

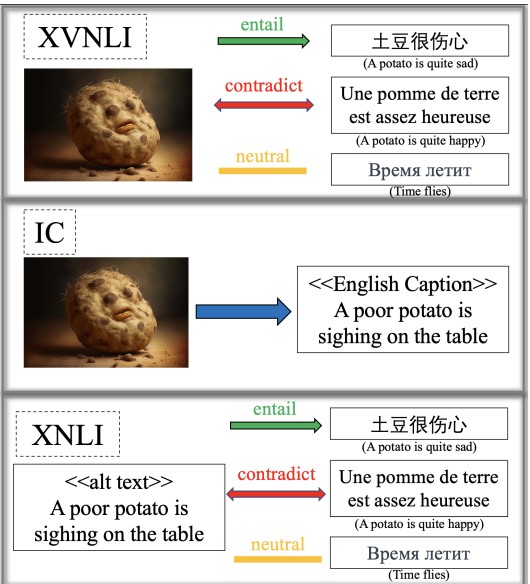

Figure 1: We employ XVNLI as a case study to exemplify the partitioning of the original task into two stages. The original task comprises an image premise and text hypothesis, which we display at the top. Below it, we present the two stages: image captioning (IC) and cross-lingual natural language inference (XNLI). The English translations of the text are provided within the brackets.

et al., 2023; Nooralahzadeh and Sennrich, 2022) have been explored to enhance cross-lingual generalization. However, there still exists a significant performance gap between English and other languages, highlighting the challenge of scarcity.

To address these challenges, this paper introduces ICU (Image Caption Understanding), which approaches V&L tasks by dividing them into two stages: image captioning (IC) and cross-lingual language understanding (XLU). As depicted in Figure 1, we use the Cross-lingual Visual Natural Language Inference (XVNLI) task as an example. Initially, we employ a V&L model to perform IC and generate an English caption for the image. This caption is then treated as the alt text for the image, enabling cross-lingual natural language infer-

---

[1]Code to reproduce our results is available at https://github.com/gjwubyron/ICU

| Frame | Template |
|-------|----------|
| 0 | $\{left\_caption\}$ $\{right\_caption\}$ |
| 1 | $< \{left\_caption\} > < \{right\_caption\} >$ |
| 2 | Left: $\{left\_caption\}$. Right: $\{right\_caption\}$. |
| 3 | Left: $< \{left\_caption\} >$. Right: $< \{right\_caption\} >$. |
| 4 | There are $\{left\_caption\}$ in the left image and $\{right\_caption\}$ in the right image. |
| 5 | The left image shows $\{left\_caption\}$ while the right image shows $\{right\_caption\}$. |

Table 1: **Hand-crafted templates.** We use direct caption concatenation in Frame0. For clarity, we enclose each caption in angle brackets in Frame1 and Frame3 as alt text. We also indicate their positions in Frame2 and Frame3. Moreover, we seamlessly integrate these captions into detailed descriptions in Frame4 and Frame5.

ence (XNLI) using a multilingual language model (mLM). ICU leverages the strengths of both the V&L model and the mLM. Given that multilingual text data are relatively more abundant and of higher quality, ICU helps alleviate the scarcity problem.

In this study, we assess our approach using two tasks from IGLUE: XVNLI and MaRVL(Liu et al., 2021), a Multicultural Reasoning over Vision and Language dataset. Our findings indicate that ICU, even in zero-shot scenarios, achieves remarkable performance on both tasks. Additionally, we observe that employing few-shot learning techniques for XVNLI further enhances the model's performance. Moreover, we explore frame engineering techniques, wherein we assign captions to different frames (refer to Table 1 for more details), and demonstrate that the model exhibits sensitivity to different frames when applied to MaRVL.

Our contributions are summarized as follows:

- We introduce ICU, an innovative divide-and-conquer approach designed to address the challenges posed by multilingual vision-and-language tasks.

- We achieve state-of-the-art results in two tasks from IGLUE benchmark, outperforming the existing multilingual methods in several languages.

- We conduct experiments and analysis to explore efficient and computationally cheap ways to further boost performance.

## 2 ICU: Image Caption Understanding

In this section, we will discuss the challenges posed by the implementation of ICU. Firstly, a crucial task is to adapt the second stage (XLU) to a suitable NLP task. For XVNLI, this can be easily addressed since NLI has already been extensively

studied. However, for MaRVL, the model needs to determine whether a textual description is true or false about a pair of images. In this case, the adaptation is achieved by assigning the two captions to different frames, as illustrated in Table 1. We then treat the task as zero-shot text classification (Yin et al., 2019). Another challenge encountered in ICU is the mLM's handling of code-switching, such as when the premise is in English while the hypothesis is in other languages. Remarkably, we demonstrate that the mLM already achieves good performance in zero-shot scenarios, and the performance can be further improved through few-shot learning.

## 3 Experiments

In this section, we will provide a comprehensive description of the models employed in ICU, along with the experimental settings and evaluations conducted.

### 3.1 Models for ICU

We use two pre-existing models for utilization in the ICU setting. For the cross-modal part, we employ OFA (Wang et al., 2022b), a sequence-to-sequence vision-and-language framework. Specifically, we select $OFA_{Large}$, which has undergone fine-tuning on COCO (Lin et al., 2015), a substantial dataset for image captioning. For decoding, $OFA_{Large}$ employs beam search with a beam size of five, while incorporating a constraint of maintaining n-gram diversity within a context window of three. For the cross-lingual part, we use mDeBERTaV3 Base (He et al., 2021), which achieves a new state-of-the-art on XNLI (Conneau et al., 2018) across 15 languages after fine-tuning. As the model is fine-tuned in a monolingual fashion (Laurer et al., 2023), meaning both the premises and hypotheses are in the same language, we continue

| Model | XVNLI | | | | | MaRVL | | | | | |
|---|---|---|---|---|---|---|---|---|---|---|---|
| | ARB | SPA | FRA | RUS | avg | IND | SWA | TAM | TUR | CMN | avg |
| mUNITER | 46.73 | 56.96 | 59.36 | 51.72 | 53.69 | 54.79 | 51.17 | 52.66 | 54.66 | 55.34 | 53.72 |
| xUNITER | 51.98 | 58.94 | 63.32 | 59.71 | 58.49 | 55.14 | 55.51 | 53.06 | 56.19 | 53.06 | 54.59 |
| $UC^2$ | 56.19 | 57.47 | **69.67** | **64.86** | **62.05** | 56.74 | 52.62 | **60.47** | 56.70 | **59.88** | **57.28** |
| $M^3P$ | 55.24 | 58.85 | 56.36 | 62.54 | 58.25 | 56.47 | **55.69** | 56.04 | 56.78 | 55.04 | 56.00 |
| ICU | **58.00** | **61.04** | 63.21 | 61.39 | 60.91 | **56.91** | 55.60 | 57.89 | **58.31** | 56.92 | 57.13 |

Table 2: **Zero-shot accuracy on XVNLI and MaRVL.** The results of the four models in the middle row are directly copied from IGLUE to enable comparison. The best performance is denoted by highlighting it in bold. (Since frame engineering is also zero-shot, we choose the best one among the frames)

to categorize it as a zero-shot application within our approach.

### 3.2 Few-shot Learning Setup

As the IGLUE benchmark does not offer comprehensive few-shot data for MaRVL, our few-shot learning efforts are solely focused on XVNLI. When conducting few-shot learning, we freeze the V&L model and exclusively adjust the parameters of the mLM. The process of freezing the V&L model can make it more efficient by enabling the reuse of captions and leveraging the significantly smaller mLM compared to the standard V&L models. Given the scarcity of few-shot data, we refrain from engaging in hyperparameter optimization, which, while potentially arbitrary, serves the purpose of safeguarding the model from overfitting on such a limited dataset. We choose to use a smaller batch size of 8, increase the learning rate to 1e-4, and limit the training to just 3 epochs. The rest hyperparameters remain the same to the fine-tuning configurations of mDeBERTaV3 (He et al., 2021). Few-shot learning is performed separately for each language.

### 3.3 Baseline Models

The models in the baseline are all initialized from mLMs, and further trained with multiple objectives to learn multimodal representations. mUNITER and xUNITER (Liu et al., 2021) expand the UNITER (Chen et al., 2020) architecture to encompass multiple languages. $M^3P$ (Ni et al., 2021) additionally introduces training tasks that involve code-switching in a multimodal context, where English caption words are randomly substituted with translations using a specific probability. $UC^2$ (Zhou et al., 2021) acquires data in five different lan-

guages with machine translation, thereby enhancing its multilingual capabilities. xUNITER, $M^3P$, and $UC^2$ all have their initializations derived from XLM-R (Conneau et al., 2020), while mUNITER is initialized from mBERT (Devlin et al., 2019). These models also differ in size, with mUNITER at 185M, xUNITER at 284M, $UC^2$ at 282M, and $M^3P$ at 377M. In contrast, the mDeBERTaV3 Base used in our approach is of a smaller size at 86M.

### 3.4 Tasks

We assess our method through two tasks. The first task, XVNLI, involves conducting inference in a multi-lingual scenario based on the image premise and text hypothesis. It comprises 357 images and 1.1k samples across 4 languages. On the other hand, MaRVL focuses on determining the truthfulness of grounded statements regarding pairs of images. It encompasses 4.9k images and 5.7k samples across 5 languages.

## 4 Results and Analysis

In this section, we present the results of ICU in comparison to existing works within the IGLUE benchmark. Additionally, we analyze the impact of few-shot learning and frame engineering techniques on the performance of ICU.

### 4.1 Overall Results

The zero-shot results for XVNLI and MaRVL are displayed in Table 2. Among the nine languages, ICU achieves the state-of-the-art (SOTA) performance in four languages, while maintaining comparable performance in the remaining languages. However, on average, it slightly lags behind the current SOTA in the IGLUE benchmark.

Figure 2: **ICU performance across different shots on XVNLI.** In our evaluation, we define one image as one shot, although typically an image may be utilized in multiple samples. On average, each shot comprises three samples.

| Model | ARB | SPA | FRA | RUS | avg |
|---|---|---|---|---|---|
| mUNITER | 46.91 | 57.73 | 59.36 | 51.80 | 53.95 |
| xUNITER | 54.04 | 60.22 | 64.52 | 63.40 | 60.55 |
| $UC^2$ | 56.87 | 62.80 | **69.76** | 65.29 | **63.68** |
| $M^3P$ | 56.01 | 60.40 | 58.59 | 62.46 | 59.37 |
| ICU | **60.70** | **64.61** | 62.61 | **65.57** | 63.37 |

Table 3: **Max-shot XVNLI results.** This evaluation is conducted under the max-shot setting, encompassing a total of 48 shots.

## 4.2 Few-shot Learning

Figure 2 illustrates the performance variations in XVNLI as the number of shots increases. Overall, a consistent upward trend can be observed, indicating an improvement in performance. Nonetheless, we observe that when the number of shots is fewer than ten, the model's performance is inferior to that of zero-shot. We hypothesize that in situations with a limited number of shots, the tuning process may result in a model with reduced generality. It's only with an adequate number of shots that the model can truly achieve noteworthy performance enhancements. Furthermore, Table 3 provides a comparison of the maximum shot performance, where ICU demonstrates a slight advantage over the previous SOTA approach in an additional language and successfully closes the performance gap on average.

## 4.3 Frame Engineering

Figure 3 depicts the performance across different frames in MaRVL. Our findings indicate that employing a simple and concise frame generally yields better results. Conversely, incorporating lengthy texts around the captions does not lead to improved performance.

## 5 Related Work

**Image-to-text Transformation in Vision-and-language Modeling** TRiG (Gao et al., 2022) and PICa (Yang et al., 2022) are two prior studies that engage in image-to-text transformation as a solution for addressing multimodal challenges in the context of visual question answering tasks. TRiG utilizes three types of transformations, encompassing image captioning, dense labeling, and optical character recognition. On the other hand, PICa employs a variety of image captioning models and tagging models to perform image transformations. Nevertheless, their efforts are concentrated exclusively on the English language.

**Vision-and-language Models** Large-scale pretraining has become the cornerstone of vision-and-language (V&L) research. Recent advancements have seen the development of big foundation models like SimVLM, Flamingo, and GIT (Wang et al., 2022c, Alayrac et al., 2022, Wang et al., 2022a). These models rely on training with sufficiently large datasets, typically constructed using image-text pairs obtained from web crawling, such as the 400 million pairs used in CLIP (Radford et al., 2021). However, due to the predominance of English in the training data, these models face challenges in effectively handling non-English inputs.

**Multilingual Language Models** The success of models like mBERT (Devlin et al., 2019) and XLM (Conneau and Lample, 2019) has demonstrated that large-scale pretraining of Transformers across multiple languages can yield impressive results in cross-lingual language understanding (XLU). With the addition of more languages and increased training data, XLM-R (Conneau et al., 2020) has surpassed mBERT's performance on various XLU benchmarks. Notably, mDeBERTaV3 (He et al., 2021) has recently achieved state-of-the-art results on XNLI, attaining a zero-shot cross-lingual accuracy of 79.8%. However, it is crucial to note that these models are primarily trained for NLP tasks and may not possess the capability to handle multimodal tasks involving both vision and

Figure 3: **ICU performance across different frames on MaRVL.** We conduct evaluation using all the frames listed in Table 1.

language.

**Multilingual Vision-and-language Models** To facilitate the learning of universal representations across different modalities and multilingual texts, the $M^3P$ framework (Ni et al., 2021) was introduced as the first pre-training framework that optimizes multiple pre-training objectives. Another unified framework, $UC^2$ (Zhou et al., 2021), proposes a novel architecture and introduces new pre-training tasks. Both $M^3P$ and $UC^2$ have demonstrated improved performance on various multilingual V&L tasks. However, there still exist noticeable performance gaps between English and non-English languages.

**Evaluation** The recently introduced IGLUE benchmark presents a new challenge for multilingual V&L models. This benchmark encompasses five tasks spanning 20 languages, thereby expanding the evaluation scope beyond previous image-text retrieval tasks such as Multi30k (Elliott et al., 2016) and MSCOCO (Lin et al., 2015).

## 6 Conclusion

In this paper, we introduce ICU, a divide-and-conquer approach designed to address the challenges of multilingual vision-and-language (V&L) tasks. ICU leverages the strengths of both V&L models and multilingual language models (mLM) to tackle the inherent difficulties in these tasks. By dividing the original tasks into two stages, we transfer the burden of multilingual processing from the V&L model to the mLM, making it a more feasible objective. This approach not only helps alleviate the scarcity problem to some extent but also proves to be more efficient.

We provide valuable insights into adapting V&L tasks to be compatible with mLMs. Furthermore, we explore the benefits of few-shot learning and

frame engineering techniques in enhancing performance. Our experimental results demonstrate the efficacy of recycling existing models, achieving state-of-the-art performance. Overall, ICU presents a promising solution for multilingual V&L tasks and opens up avenues for future research.

## Limitations

While our study focuses on exploring adaptations for two specific V&L tasks, it is important to acknowledge that the adaptation process can be challenging for other tasks. Take xGQA (Pfeiffer et al., 2022) as an example, it can not be easily converted to a Question Answering task, since the caption are usually too short to include the whole context of the image. The scarcity problem, particularly prevalent in low-resource languages like Tamil in the MaRVL dataset, continues to persist.

## Acknowledgments

We would like to thank Dr. Farhad Nooralahzadeh for the comprehensive seminar course in Multimodal Multilingual Natural Langauge Processing, Emanuele Bugliarello for the explanation regarding the dataset, and the anonymous reviewers for their valuable comments and feedbacks.

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
