# OpenReview forum: "ICU: Conquering Language Barriers in Vision-and-Language Modeling by Dividing the Tasks into Image Captioning and Language Understanding"
_EMNLP/2023/Conference — EMNLP 2023 Findings_

### Official Review · Reviewer_UdNy · 2023-07-20

**Soundness:** 3

**Excitement:**

3: Ambivalent: It has merits (e.g., it reports state-of-the-art results, the idea is nice), but there are key weaknesses (e.g., it describes incremental work), and it can significantly benefit from another round of revision. However, I won't object to accepting it if my co-reviewers champion it.

**Paper Topic And Main Contributions:**

The paper present an approach for image captioning understanding (ICU) by decomposing a single vision-to-language model into two models: one for visual understanding and another for multilingual language understanding. They make use of vanilla pre-trained models for each decomposed tasks, and report results that beat some SOTA results on the IGLUE benchmark.

**Questions For The Authors:**

A) Has something similar been proposed for tasks beyond XNLI?

**Reasons To Accept:**

The approach is simple and works well, and that should should be praised in my opinion
They report new SOTA results in multilingual vision understanding tasks

**Reasons To Reject:**

I'm not so sure that the vision part is interesting for this community, so maybe a more general conference such as NeurIPS or AAAI might be more suitable for this type of paper. And there is no contribution in the language aspects
I know it is a short paper, but it would benefit from a clearer explanation of the tasks/datasets and the methods

**Reproducibility:**

4: Could mostly reproduce the results, but there may be some variation because of sample variance or minor variations in their interpretation of the protocol or method.

**Reviewer Confidence:**

3: Pretty sure, but there's a chance I missed something. Although I have a good feel for this area in general, I did not carefully check the paper's details, e.g., the math, experimental design, or novelty.

**Typos Grammar Style And Presentation Improvements:**

It would be nice to mention which metric is used in the results reported in Table 2. I imagine it is accuracy, but I'm not sure

---

> ### Author Rebuttal · Authors · 2023-08-27
>
> ## Reasons To Reject:
> **I'm not so sure that the vision part is interesting for this community, so maybe a more general conference such as NeurIPS or AAAI might be more suitable for this type of paper. And there is no contribution in the language aspects I know it is a short paper, but it would benefit from a clearer explanation of the tasks/datasets and the methods**
>
> **Response:** While we understand the shared concern regarding the vision component, we remain confident that our work is a suitable fit for this community, given the undeniable significance of vision as a pivotal modality. An illustrative example is the MaRVL dataset employed in our research, which not only achieved acceptance at EMNLP21 but also secured the esteemed recognition of the best paper award.
>
> Furthermore, we'd like to emphasize that our approach introduces a novel challenge within the realm of language aspects. Unlike previous works in the context of XNLI, where both the premise and hypothesis were consistently in the same languages, our approach introduces a distinctive setting where the premise is in English, and the hypothesis is in other languages. Therefore, it's important to recognize that our work does indeed make a meaningful contribution in this dimension. We deeply regret any confusion that may have arisen due to our earlier lack of clarity in explaining these aspects.
>
> ## Questions For The Authors:
>
> A) **Has something similar been proposed for tasks beyond XNLI?**
>
> **Response:** We ventured into applying our method to another task, namely xGQA. However, as outlined in the limitations section, adapting our approach to this task posed considerable challenges. xGQA, at its core, is a classification task characterized by an extensive label space exceeding 1800 labels. Consequently, conventional NLP techniques such as text classification (typically effective with fewer than 10 labels), question answering (often used to extract answers from lengthy paragraphs), or text generation (which poses open domain challenges) do not seamlessly align as downstream models for processing the captions we obtain.
>
> ## Typos Grammar Style And Presentation Improvements:
>
> **Response:** Indeed, that detail slipped our notice. We appreciate your pointing it out, and we indeed employed accuracy as the metric. Thank you for raising this concern.

---

### Official Review · Reviewer_ZPdR · 2023-08-01

**Typos Grammar Style And Presentation Improvements:** N.A
**Soundness:** 3

**Excitement:**

2: Mediocre: This paper makes marginal contributions (vs non-contemporaneous work), so I would rather not see it in the conference.

**Missing References:**

N.A

**Paper Topic And Main Contributions:**

- What is this paper about? This paper aims to solve the multilingual vision and language task.
- what contributions does it make? This paper proposes to divide the task into two-stage: Image captioning in English and using the English text for later multilingual understanding.

**Questions For The Authors:**

N.A

**Reasons To Accept:**

- This simple idea gives state-of-the-art results on partial tasks of XVNLI and MaRVL of multilingual vision and language tasks.
- With better training on the sub-module (mLM), the proposed model can give better performance.
- The idea to divide the multilingual vision and language task into Image captioning and multilingual understanding is intuitive.

**Reasons To Reject:**

- The chained impacts of image captioning and multilingual understanding model in the proposed pipeline. If the Image caption gives worse results and the final results could be worse. So The basic performance of the image caption model and multilingual language mode depends on the engineering model choice when it applies to zero-shot.
- This pipeline style method including two models does not give better average results for both XVNLI and MaRVL. Baseline models in the experiments are not well introduced.

**Reproducibility:**

3: Could reproduce the results with some difficulty. The settings of parameters are underspecified or subjectively determined; the training/evaluation data are not widely available.

**Reviewer Confidence:**

4: Quite sure. I tried to check the important points carefully. It's unlikely, though conceivable, that I missed something that should affect my ratings.

---

> ### Author Rebuttal · Authors · 2023-08-27
>
> ## Reasons To Reject:
>
> 1. **The chained impacts of image captioning and multilingual understanding model in the proposed pipeline. If the Image caption gives worse results and the final results could be worse. So The basic performance of the image caption model and multilingual language mode depends on the engineering model choice when it applies to zero-shot.**
>
> **Response:** We concur that the cascading impact is a valid consideration. This is precisely why we've opted for a state-of-the-art model that delivers high-quality captions effectively even in a zero-shot scenario. Additionally, our multilingual language model has showcased remarkable zero-shot performance, as demonstrated in Table 2.
>
> In support of our methodology, the engineering aspect is considerably less demanding in comparison to the training processes. Hence, experimenting with various options could prove beneficial without significant effort.
>
> 2. **This pipeline style method including two models does not give better average results for both XVNLI and MaRVL. Baseline models in the experiments are not well introduced.**
>
> **Response:** Our approach may not present superior average results. However, it's crucial to highlight that it does yield enhanced performance in four or five (including the max-shot scenario) out of the nine languages considered.
>
> Absolutely, we should have provided more comprehensive insights into the baseline models. However, we'd like to emphasize a crucial aspect here. Our method boasts adjustable parameters of 86M (specifically, mDeBERTaV3 base), whereas the competing models have 185M for mUNITER, 284M for xUNITER, 282M for UC$^2$, and 377M for M$^3$P. In practical terms, our method is more parameter-efficient, and it's imperative not to underestimate the performance it can deliver.
>
> Furthermore, our approach introduces the opportunity to reuse an already trained image captioning model, a practice that undoubtedly aligns with ecological considerations. This stands in contrast to training a model from scratch, as done in the baseline, which is inherently more resource-intensive.

---

### Official Review · Reviewer_oSwq · 2023-08-03

**Typos Grammar Style And Presentation Improvements:** 1. In Figure 1, English translations …
**Soundness:** 3

**Excitement:**

3: Ambivalent: It has merits (e.g., it reports state-of-the-art results, the idea is nice), but there are key weaknesses (e.g., it describes incremental work), and it can significantly benefit from another round of revision. However, I won't object to accepting it if my co-reviewers champion it.

**Missing References:**

The following works also propose generating and using image captions as an intermediate clue to solve a VQA task while it is not a multilingual task. Nevertheless, the idea is quite similar to the proposed method in this paper and these works should be acknowledged and referred to in the paper.

- Gao, Feng, et al. "Transform-Retrieve-Generate: Natural Language-Centric Outside-Knowledge Visual Question Answering." Proceedings of the IEEE/CVF Conference on Computer Vision and Pattern Recognition. 2022.
- Yang, Zhengyuan, et al. "An Empirical Study of GPT-3 for Few-Shot Knowledge-Based VQA." Proceedings of the AAAI Conference on Artificial Intelligence. 2022.

**Paper Topic And Main Contributions:**

This paper proposes a multilingual multimodal method that divides a task into two sub-tasks: English caption generation and cross-lingual natural language understanding. The paper evaluates the proposed method on two tasks, XVNLI and MaRVL, in nine languages. Experimental results show that the proposed method outperforms existing strong baselines in four languages while the results are comparable with the baselines in the rest of the languages.

**Questions For The Authors:**

A. How are image captions generated from OFA-large? Generating captions involves several hyper-parameters such as the decoding algorithm (i.e., greedy or beam search), beam size if beam search is used, penalty factor on repetitive sequence, etc.

B. How is the proposed method evaluated in the zero-shot setting? Table 2 presents the zero-shot performance while the proposed method, especially mDeBERTaV3, is trained in the downstream tasks as described in Lines 138 - 139. Thus, it is unclear what the zero-shot performance means and how it is computed.

C. How is the proposed method trained in the downstream tasks? The datasets used in the paper are multilingual multimodal ones, and thus, there would be two ways of training; training a method jointly on all the languages and evaluating individually on each language, or training individually on each language and then evaluating in the same manner. Detailed experimental setup should be described for reproducibility.

D. In Figure 2, why is the performance with few-shot that is less than 10 consistently worse than that of zero-shot in all the languages?

**Reasons To Accept:**

1. The proposed idea and method are straightforward and effective.
2. The paper is easy to follow.

**Reasons To Reject:**

1. The paper lacks the details of the implementation of the proposed method as well as the experimental setup, which makes it difficult to reproduce the experiments. See the question section below.
2. Citation of the related work to the idea of the proposed method is missing. See the missing reference section below.

**Reproducibility:**

4: Could mostly reproduce the results, but there may be some variation because of sample variance or minor variations in their interpretation of the protocol or method.

**Reviewer Confidence:**

4: Quite sure. I tried to check the important points carefully. It's unlikely, though conceivable, that I missed something that should affect my ratings.

---

> ### Author Rebuttal · Authors · 2023-08-27
>
> ## Reasons To Reject:
> 1. **The paper lacks the details of the implementation of the proposed method as well as the experimental setup, which makes it difficult to reproduce the experiments. See the question section below.**
>
>
> **Response:** We understand the importance of clarity and completeness in ensuring experiment reproducibility. However, our intricate setup, involving multiple data sources, two distinct models (Image Captioning and language tasks), can result in lengthy explanations. To address this, we've chosen to provide detailed instructions in our supplementary README file, available within the submitted zip file. This approach maintains manuscript readability while facilitating easy replication. We will make our code repository publically available upon paper acceptance.
>
> 2. **Citation of the related work to the idea of the proposed method is missing. See the missing reference section below.**
>
> **Response:** Upon a cursory examination of those papers, we find ourselves in complete agreement on this matter. Your reminder is greatly appreciated.
>
> ## Questions For The Authors:
> A. **How are image captions generated from OFA-large? Generating captions involves several hyper-parameters such as the decoding algorithm (i.e., greedy or beam search), beam size if beam search is used, penalty factor on repetitive sequence, etc.**
>
> **Response:** Regarding hyperparameters, we have adopted identical settings to those outlined in the OFA paper to which we refer. Specifically, our decoding algorithm employs beam search with a beam size of five, complemented by a penalty factor of 3.
>
> B. **How is the proposed method evaluated in the zero-shot setting? Table 2 presents the zero-shot performance while the proposed method, especially mDeBERTaV3, is trained in the downstream tasks as described in Lines 138 - 139. Thus, it is unclear what the zero-shot performance means and how it is computed.**
>
> **Response:** We apologize for any confusion caused by the lack of clarity in our initial manuscript. Throughout this study, our performance metric is accuracy. The term "zero-shot" indicates that mDeBERTaV3 has not been fine-tuned for this specific task, where the premise is in English while the hypothesis is presented in other languages. The zero-shot results are listed in Table 2.
>
> Lines 138-139 delineate our approach in the few-shot scenario, involving the fine-tuning of mDeBERTaV3 with varying shots. The collective few-shot outcomes are visually depicted in Figure 2, while specific results for the maximum shots are summarized in Table 3.
>
> C. **How is the proposed method trained in the downstream tasks? The datasets used in the paper are multilingual multimodal ones, and thus, there would be two ways of training; training a method jointly on all the languages and evaluating individually on each language, or training individually on each language and then evaluating in the same manner. Detailed experimental setup should be described for reproducibility.**
>
> **Response:** Indeed, this aspect escaped our attention. We pursued the second approach, whereby we conducted separate training for each language and evaluated them individually.
>
> D. **In Figure 2, why is the performance with few-shot that is less than 10 consistently worse than that of zero-shot in all the languages?**
>
> **Response:** We hypothesize that in situations with a limited number of shots, the tuning process may result in a model with reduced generality. It's only with an adequate number of shots that the model can truly achieve noteworthy performance enhancements.
>
> ## Typos Grammar Style And Presentation Improvements:
>
> **Response:** We are genuinely appreciative of your thoughtful and valuable feedback, as they are certain to contribute to the improvement of our manuscript.

---

### Meta-Review · Area_Chair_xf7X · 2023-09-20

**Recommendation:** 3

**Metareview:**

The paper proposes a method that splits multilingual multimodal understanding tasks into two stages:  English caption generation and cross-lingual natural language understanding.

Pros:  The proposed approach is intuitive and achieves SOTA results in the part of the experimental studies

Cons: The method is hard to generalize for tasks beyond XNLI due to this two stage setup. Also the method does not give better average results for both XVNLI and MaRVL

Additionally the draft needs to be updated with more experimental details like use of single-language at a time modeling, baselines and other details shared during the author discussion period.

---

### Decision · Program_Chairs · 2023-10-07

**Decision:**

Accept-Findings

**Comment:**

The paper proposes a method that splits multilingual multimodal understanding tasks into two stages:  English caption generation and cross-lingual natural language understanding.

Pros:  The proposed approach is intuitive and achieves SOTA results in the part of the experimental studies

Cons: The method is hard to generalize for tasks beyond XNLI due to this two stage setup. Also the method does not give better average results for both XVNLI and MaRVL

Additionally the draft needs to be updated with more experimental details like use of single-language at a time modeling, baselines and other details shared during the author discussion period.